# Muscle Wasting in Chronic Kidney Disease: Mechanism and Clinical Implications—A Narrative Review

**DOI:** 10.3390/ijms23116047

**Published:** 2022-05-27

**Authors:** Tsai-Chin Cheng, Shou-Hsien Huang, Chung-Lan Kao, Po-Cheng Hsu

**Affiliations:** 1Department of Physical Medicine and Rehabilitation, Taipei Veterans General Hospital, Taipei 112, Taiwan; shps951023@gmail.com (T.-C.C.); clkao@vghtpe.gov.tw (C.-L.K.); 2Department of Physical Medicine and Rehabilitation, Far Eastern Memorial Hospital, New Taipei City 220, Taiwan; mcgrady152002@hotmail.com; 3School of Medicine, National Yang-Ming Chiao-Tung University, Taipei 112, Taiwan

**Keywords:** chronic kidney disease, muscle wasting, molecular mechanism, protein metabolism, muscle regeneration, exercise, nutrition, physical modality, pharmaceutical intervention

## Abstract

Muscle wasting, known to develop in patients with chronic kidney disease (CKD), is a deleterious consequence of numerous complications associated with deteriorated renal function. Muscle wasting in CKD mainly involves dysregulated muscle protein metabolism and impaired muscle cell regeneration. In this narrative review, we discuss the cardinal role of the insulin-like growth factor 1 and myostatin signaling pathways, which have been extensively investigated using animal and human studies, as well as the emerging concepts in microRNA- and gut microbiota-mediated regulation of muscle mass and myogenesis. To ameliorate muscle loss, therapeutic strategies, including nutritional support, exercise programs, pharmacological interventions, and physical modalities, are being increasingly developed based on advances in understanding its underlying pathophysiology.

## 1. Introduction

As renal disease progresses, the deteriorated renal function and chronic uremic inflammatory status lead to nutritional and metabolic alterations. Protein-energy wasting (PEW) syndrome, defined as a simultaneous loss of body protein and energy stores, frequently develops and contributes to muscle wasting in patients with chronic kidney disease (CKD) [1]. Progressively, loss of muscle protein causes a decline in muscle strength and physical performance, which is clinically defined as sarcopenia [2]. The prevalence of sarcopenia in CKD ranges from 4 to 42%, based on different definitions, study populations, or CKD stages, which is reportedly higher in patients undergoing dialysis, ranging between 13.7 and 42.2% [3]. Current evidence has shown that several complications associated with CKD contribute to dysregulated muscle protein balance, including anemia, metabolic acidosis, abnormal growth hormone axis and androgen deficiencies, insulin resistance, inflammation, hemodialysis (HD) process, and disuse [4], exhibiting varying extents from early CKD to dialysis [5]. Other studies have indicated that CKD-related mineral and bone disorders also involve in the development of muscle wasting in CKD [6], including vitamin D deficiency [7] and secondary hyperparathyroidism [8]. The complex interactions of these CKD factors and pathophysiological changes at the cellular level have been explored [9,10,11], but remain poorly understood.

PEW, muscle wasting, and sarcopenia strongly correlate with frailty and ultimately a reduction in health-related quality of life and an increased risk of hospitalization, morbidity, and mortality in patients with CKD [3,12,13]. To develop effective treatments that can counter the pathophysiological changes in CKD and ameliorate the loss of muscle mass and function, it is necessary to advance our existing knowledge regarding the underlying molecular pathways of muscle wasting in CKD. In this narrative review, we highlighted mechanistic insights into the cellular signaling pathways that contribute to muscle wasting in CKD. In addition, we summarized existing therapeutic strategies and future directions to suppress or attenuate progressive muscle protein loss in CKD.

## 2. Mechanism Underlying Muscle Wasting in CKD

Previously, muscle wasting in CKD has frequently been attributed to malnutrition due to the common presentation of hypoalbuminemia. Malnutrition describes the consequences of inadequate food intake, possibly associated with PEW syndrome in CKD [14]. However, one study has shown that reduced serum albumin levels were more significantly related to inflammation than protein intake [15]. Another study by Pupim et al. demonstrated that increasing protein intake could improve protein homeostasis in patients undergoing chronic HD; however, the effects are only transient and return to baseline levels two hours after one HD session [16]. The findings indicate that muscle loss in CKD contributes to the underlying catabolic mechanisms associated with renal impairment, which cannot simply be attributed to malnutrition [17,18,19].

In healthy adults, whole-body proteins are in a dynamic state, undergo degradation to amino acids, and are replaced by the synthesis of new proteins. The daily turnover rate is estimated to be 250–300 g in a 60-kg man, of which approximately 100–120 g is from skeletal muscle [20]. Constant muscle mass is maintained through strictly regulated protein synthesis and degradation, and small, but persistent, changes in the balance of protein metabolism can lead to substantial muscle wasting over time. CKD-associated muscle wasting results from an altered balance between catabolic and anabolic processes that control muscle homeostasis. The underlying molecular pathways that disrupt muscle growth and turnover mainly involve dysregulated protein metabolism (increased protein degradation and suppressed protein synthesis) and impaired muscle regeneration [21,22] (Figure 1).

Specific cellular signaling pathways responsible for muscle homeostasis have been identified, including the insulin-like growth factor-1(IGF-1) pathway, which promotes protein synthesis and recruitment of muscle satellite cells, and the myostatin pathway, which causes muscle protein degradation and inhibits satellite cell function [23] (Figure 2). A few studies have investigated the clinical applications of IGF-1 and myostatin as potential biomarkers of muscle wasting. Among patients receiving HD, serum IGF-1 levels were found negatively correlated with sarcopenia, whereas myostatin levels correlated positively with sarcopenia [24]. Similarly, Han et al. revealed that a higher serum myostatin level significantly increased the possibility of low grip strength in patients with CKD [25]. These results indicate that catabolic conditions in CKD induce an imbalance between these two closely interacting pathways, leading to an accelerated rate of muscle wasting [26,27]. Recently, a growing body of evidence has focused on the involvement of microRNAs (miRNAs) [28,29,30] and gut microbiota [31,32,33] in muscle wasting. Alterations in miRNA expression in CKD can lead to abnormal protein metabolism and muscle regeneration through the modulatory effects of miRNAs on relevant intracellular signaling pathways [22]. Disruption of gut microbiota homeostasis is increasingly recognized to play a critical role in inflammation, insulin resistance, and mitochondrial dysfunction, which are closely associated with muscle wasting in CKD [34,35].

### 2.1. Increased Protein Degradation

Several cellular mechanisms that contribute to increased protein degradation in CKD have been proposed, including the activation of the ubiquitin (Ub)-proteasome system (UPS), caspase-3, and autophagy by lysosome [9,36].

#### 2.1.1. Activation of the Ub-Proteasome System and Caspase-3

The Ub-proteasome system (UPS) has been recognized as a major pathway degrading protein in skeletal muscles [37]. Protein degradation is a well-controlled process, including the activation of Ub by E1 (Ub-activating enzyme) and E2 (Ub-carrier proteins) enzymes and the formation of the E3-protein complex by a specific E3 enzyme (Ub-protein ligase) and targeted protein substrate [38]. This conjugation process is repeated until a Ub chain containing five Ubs is formed and transported to the 26S proteasome, which cleaves the protein substrate into small peptides [27,39]. UPS activation is initiated by the impaired IGF-1-Phosphoinositide 3-kinases(PI3K)-Protein kinase B(Akt) signaling pathway developed in CKD with insulin resistance, metabolic acidosis, excessive angiotensin II, and inflammation [27,40]. The IGF-1-PI3K-Akt signaling pathway is a major determinant of skeletal muscle homeostasis by promoting protein synthesis via mammalian target of rapamycin (mTOR) signaling and inhibiting protein degradation through inactivation of the Forkhead box O (FoxO) transcription factor [36]. Defective IGF-1-PI3K-Akt signaling activates the FoxO transcription factor, which permits its nuclear translocation, stimulating the expression of atrogenes such as *atrogin-1/muscle atrophy F-box (MAFbx)* and *muscle ring factor 1 (MuRF-1)* (Figure 2). These enzymes belong to the muscle-specific E3 Ub ligase family, which recognizes specific muscle proteins and facilitates proteolysis via UPS [21,27,36].

On the other hand, the myostatin signaling pathway is a critical negative regulator of skeletal muscle mass, which is upregulated in skeletal muscle of patients with CKD [41] in response to oxidative stress, inflammation, uremic toxins, angiotensin II, glucocorticoids, and metabolic acidosis in CKD [26,42]. Myostatin or growth development factor-8 (GDF-8) is a member of the transforming growth factor-β (TGF-β) superfamily and is secreted by mature muscle cells [43]. The role of myostatin in muscle wasting during CKD involves UPS activation and inhibition of muscle satellite cell recruitment [26]. At the muscle cell surface, the binding of myostatin to the activin type IIB receptor (ActRIIB) leads to activation of mothers against decapentaplegic homolog 2/3 (Smad2/Smad3) transcription factors, which augments the transcription of atrophy-related genes and downregulation of genes responsible for myogenesis [44]. Additionally, there exists a crosstalk between myostatin and IGF-1 signaling pathway. Both myostatin-ActRIIB binding and Smad2/Smad3 activation result in phosphorylation of Akt (reduced Akt activity) in muscle cells, decreasing FoxO phosphorylation and increasing the transcription of *atrogin*-1 and *MuRF*-1, thereby stimulating UPS-mediated protein degradation [26,44] (Figure 2).

It has been shown that muscle wasting in CKD is primarily associated with accelerated muscle proteolysis by activation of the UPS [37]. However, previous studies have found that although the UPS rapidly degrades actin or myosin, it cannot break down larger and more complex structures such as actomyosin or myofibrils [11]. Du et al. discovered that under catabolic conditions, activation of caspase-3 is an initial step that cleaves actomyosin or myofibrils into substrates that are rapidly degraded by the UPS [45]. This initial proteolytic process produces a characteristic footprint, a cleaved 14-kDa actin fragment [45], elevated in catabolic conditions, including in patients with hip arthroplasty, HD, or burn injuries [46]. These results suggest that the 14-kDa actin fragment may serve as a marker or potential diagnostic tool for muscle wasting [11]. In addition, caspase-3 stimulates protein degradation by directly stimulating proteasomal activity. This process involves the cleavage of specific subunits of the 19S proteasome by caspase-3, which alters its conformation and permits the insertion of additional protein substrates into the proteolysis site of the proteasome [47]. These caspase-3-mediated actions exhibit feed-forward amplification that augments muscle protein degradation under catabolic conditions [47].

#### 2.1.2. Lysosome-Mediated Autophagy

Autophagy is a homeostatic proteolytic process responsible for the degradation and recycling, mediated via the lysosomal machinery, of long-lived proteins and organelles, such as mitochondria and sarcoplasmic reticulum [48]. The autophagy process begins with the formation of autophagosomes on cytoplasmic targets, resulting in mature autophagosomes with cytoplasmic cargo inside, followed by the attachment of LC3-II, which further fuses with lysosomes to form degradative vesicles known as autophagolysosomes [21].

As mentioned above, FoxO transcription factors can stimulate the transcription of atrogenes, leading to UPS activation. Several studies have also found that FoxO can activate autophagy, with evidence showing stimulation of various autophagy-related genes [23,49,50]. FoxO coordinately activates protein degradation through proteasomal and autophagic pathways, thereby contributing to muscle atrophy [49]. Likewise, the myostatin signaling pathway is associated with the activation of the autophagy-lysosome process by suppressing Akt-FoxO signaling transduction [26].

Therefore, under catabolic conditions such as CKD, defective IGF-1-PI3K-Akt signaling can activate the autophagy-lysosome system by disinhibiting FoxO activity. A study conducted by Wang et al. revealed that myostatin expression is markedly upregulated in a CKD mouse model, associated with increased autophagosome formation and Ub ligase expression in mice [51]. Su et al. reported that the levels of autophagy-related proteins, autophagosome formation, and autophagosome-mediated degradation were increased in mice with CKD [52]. However, their results indicated that autophagy activation does not directly cause the breakdown of myofibrillar proteins but, more importantly, leads to deterioration of mitochondrial function and decreased ATP production [52].

Taken together, proteasomal (UPS and caspase-3) and autophagic-lysosomal mechanisms work coordinately to accelerate the process of muscle protein degradation in patients with CKD [21,53].

### 2.2. Supressed Protein Synthesis

Studies investigating the effects of CKD on muscle protein metabolism have focused on protein degradation more than protein synthesis. Some authors have claimed that CKD-induced activation of protein degradation is a more prominent cause of muscle wasting than decreased protein synthesis [9,54]. Although abundant evidence supports upregulated protein degradation in CKD-associated muscle wasting, whether protein synthesis is suppressed remains inconclusive [55,56,57]. Adey et al. showed significantly lower synthetic rates of mixed muscle proteins, myosin heavy chain, and mitochondrial proteins in patients with CKD than in healthy controls, implying that reduced protein synthesis contributes to muscle loss in patients with CKD [55].

Indeed, several CKD-related factors can reportedly attenuate protein synthesis, including metabolic acidosis, upregulated pro-inflammatory cytokine expression, and anorexia-mediated malnutrition [58]. These factors are all associated with the suppression of the IGF-1-PI3K-Akt signaling pathway [58], resulting in the inhibition of mTOR and protein synthesis, along with accelerated proteolysis that lead to muscle wasting [23]. In addition, myostatin is involved in the suppression of protein synthesis through inhibition of the mTOR pathway [26]. A key component in protein synthesis is ribosome biogenesis, which is responsible for mRNA translation to produce new muscle proteins. Emerging evidence has shown that ribosome biogenesis is critical for regulating muscle mass in response to both anabolic and catabolic stimuli through the mTOR, Wingless-related integration site (Wnt)/β-catenin/c-myelocytomatosis oncogene (c-myc), and AMP-activated protein kinase (AMPK) signaling pathways [59,60]. Altered mTOR signaling observed in CKD leads to impaired ribosome biogenesis and the subsequent suppression of protein synthesis [21]. Recently, Zhang et al. discovered a new epigenetic regulation of muscle protein synthesis through nucleolar protein 66 (NO66)-mediated suppression of ribosomal biogenesis via demethylation [61]. The authors also revealed that the expression of NO66 was upregulated in CKD mice by activating the nuclear factor kappa B (NF-kB) signaling pathway via CKD-associated inflammatory cytokines [61].

### 2.3. Impaired Muscle Regeneration

Skeletal muscle regeneration is a highly regulated process mediated by a group of specialized muscle stem cells, also known as satellite cells, located beneath the basal lamina of myofibers [62]. Satellite cells remain in a quiescent state during rest but can re-enter the cell cycle quickly upon injury or growth signals. Activated satellite cells facilitate muscle growth and regeneration via several stages of myogenesis, including self-renewal or proliferation of myoblasts, differentiation into myocytes, and fusion in the form of myotubes or myofibrils [63,64]. Myogenesis is characterized by the sequential gene expression of several myogenic regulatory factors, such as *paired-box protein-7 (Pax7)*, *myogenic factor 5 (Myf5)*, and *myoblast determination protein (MyoD)* [63]. Previous studies have discovered various extracellular signals released from the damaged muscle that contribute to satellite cell activation from quiescent to a proliferative state, including IGF-1, fibroblast growth factor (FGF), hepatocyte growth factor, nitric oxide (NO), and sphingosine-1-phosphate (S1P) [63,65].

In addition to abnormal protein metabolism, emerging evidence has shown that muscle wasting in CKD may result from an impaired muscle regeneration capacity, which is associated with dysfunctional satellite cells, and proposed mechanisms include inflammation, insulin resistance, altered myostatin, and IGF-1 signaling [9,21,64]. In a CKD mouse model, Zhang et al. found that isolated satellite cells exhibit decreased *MyoD* and *myogenin* expression, associated with impaired IGF-1 signaling in CKD. In addition, the authors found increased TGF-β1 expression and collagen deposition in regenerating muscle, indicating that impaired IGF-1 signaling in CKD leads to satellite cell dysfunction and muscle fibrosis [66]. Myostatin is another regulator of muscle regeneration by downregulation of myogenesis genes expression, via mitogen-activated protein kinases (MAPKs) and Smad2/Smad3 pathways [44] (Figure 2). Previous studies have shown that myostatin can inhibit myoblast proliferation [67] and differentiation [68], and block the activation and self-renewal of satellite cells [69]. Therefore, upregulated myostatin signaling in CKD negatively affects both muscle protein metabolism and satellite cell function, thereby causing muscle wasting [9,26,42].

### 2.4. Alterations in microRNAs Expression

MicroRNAs (miRNAs) are small non-coding RNAs that regulate gene expression by binding to the 3′ untranslated region of their target mRNA in a post-transcriptional manner, involving mRNA degradation or suppression of translation [30]. More than thousands of miRNAs have been discovered to exhibit pivotal roles in various biological processes, and aberrant miRNA expression has been detected in several diseases [70]. miRNAs are increasingly being recognized as essential regulators of muscle homeostasis through interactions with intracellular signaling pathways that regulate muscle growth, regeneration, and metabolism [22,29,30,71].

Alterations in miRNAs have been implicated in the development of muscle wasting in CKD [22], mediated via increased protein degradation [72,73,74] or impaired myogenesis [75]. Researchers have found that these miRNAs typically target multiple proteins that function in the same signaling pathway, especially the IGF-1-PI3K-Akt and myostatin pathways, closely related to muscle wasting [9,22]. Specifically, expression levels of miRNA-486 (inactivates phosphatase and tensin homolog [PTEN] and FoxO) [73], miRNA-26a (inhibits FoxO and PTEN), [74] miRNA-23a (suppresses PTEN, *MuRF-1*, and *atrogin-1*) [72], and miRNA-27a (inhibits FoxO and reduces myostatin levels) [72] are found to be substantially decreased in CKD, leading to enhanced protein degradation. Decreased miRNA-29 expression was detected in CKD mice, causing upregulation of Yin Yang-1 (YY1) transcription factor activity and subsequent inhibition of myoblast differentiation [75]. Conversely, overexpression of miRNA-23a and miRNA-27a decreases the activation of caspase-3 and -7 and increases markers of muscle regeneration [72]. miRNA-206 and -486 also facilitate myoblast differentiation by downregulating *Pax7* [76]. In summary, these experimental findings advance our understanding of the role of miRNAs in muscle wasting induced by catabolic factors in CKD and provide promising implications for therapeutic targets.

### 2.5. Gut Microbiota Dysbiosis

The human gut microbiota is composed of more than 100 trillion microorganisms in a symbiotic and commensal relationship with the host, exhibiting various regulatory effects on health maintenance, including nutrient metabolism, immune modulation, intestinal mucosal integrity and endocrine function, resistance to pathogens, energy homeostasis, biosynthesis of vitamins, hormones, and neurotransmitters, and xenobiotic detoxification [33,77]. Normally, gut microbiota remains in a homeostatic state, but its composition and diversity may evolve dynamically throughout an individual’s lifetime, depending on several factors, such as diet, age, host genes, antibiotic exposure, exercise, and comorbidities [33,77]. Gut microbiota dysbiosis refers to alterations in the composition and function of the intestinal microbial community, which are associated with pathological conditions [78]. Disrupted equilibrium in gut microbiota results in failure of gut barrier function, leading to increased intestinal permeability and an immune-mediated inflammatory response [79]. Accumulating evidence has shown that gut microbiota dysbiosis can be associated with various diseases apart from localized gastrointestinal illnesses, including cardiovascular, metabolic, hepatic, respiratory, neurologic, and oncologic disorders [33,77]. Consequently, the concept of the gut-muscle axis has been proposed to describe the remote effects of gut microbiota-derived metabolites on muscle metabolism [31,32,33]. The causal relationship between gut dysbiosis and muscle wasting is mediated by the reduced bioavailability of dietary amino acids, chronic systemic inflammation, insulin resistance, mitochondrial dysfunction, and modulation of host gene expression [31,32,33,80]. Animal studies have uncovered potential molecular pathways, including activation of PI3K-Akt, Toll-like receptor (TLRs)-NF-kB, MAPKs, and AMPK pathways, inducing upregulation of *atrogin-1* and *MuRF-1* by microbiota-derived metabolites, such as lipopolysaccharides and indoxyl sulfate, through gut barrier breakdown [81,82]. Enhanced branched-chain amino acid catabolism mediated via activated transcription factor kruppel like factor 15 (KLF15) and downstream *branched chain amino acid transaminase 2 (Bcat2)* and *branched-chain α-keto acid dehydrogenase (Bckdh)* genes, as well as impaired neuromuscular junctions (NMJs) evidenced by reduced serum choline and expression of receptor associated protein of the synapse (Rapsyn) and low-density lipoprotein receptor-related protein 4 (Lrp4) (both are important for NMJs assembly), have been reported in a germ-free mouse study [83].

In patients with CKD, gut microbiota dysbiosis has been recognized by a significant difference in gut microbiota abundance and composition when compared with healthy controls [84]. Several studies have suggested that the accumulation of uremic toxins affects the growth of commensal bacteria, contributing to an imbalanced gut microbiota [34,84]. Uremic dysbiosis may induce muscle wasting in CKD through gut-muscle crosstalk. Uchiyama et al. developed a germ-free mouse model treated with fecal microbiota transplantation obtained from either control or CKD mice, to explore the contribution of uremic dysbiosis to sarcopenia [35]. The authors demonstrated that uremic dysbiosis could induce microinflammation by facilitating intestinal permeability and, in turn, increase the level of serum uremic solutes and additionally increase bacterial fermentation product levels, ultimately resulting in sarcopenia owing to insulin resistance and mitochondrial dysfunction [35].

## 3. Therapeutic Strategies for Muscle Wasting

Emerging therapeutic strategies have been proposed to target muscle wasting in patients with CKD. In addition, considering the growing CKD population, greater awareness of holistic care has been promoted as “Renal Rehabilitation”. Notably, targeting muscle loss and related impairment of activities of daily living is an important issue, and standard-care interventions and potential therapeutics are listed in Table 1.

**Table 1 ijms-23-06047-t001:** Standard care and potential intervention against muscle wasting in patients with CKD.

Major Category	Subitem	Reference
Nutritional intervention	Dietary modification (Low protein diet or very low protein diet)	[85,86,87]
	Vitamin D supplement	[88]
	Probiotics/Prebiotics	[34,84]
	AST-120	[89,90,91]
Pharmacological therapy and potential therapeutic intervention	Myostatin inhibitors * (ActRIIB receptor blockade)	[92,93]
	Interleukin-1 blockade *	[94,95]
	Ghrelin and leptin regulation *	[96,97]
	miRNAs *	[72]
	Activin A blockade *	[98]
Exercise	Aerobic and resistance exercises	[99,100,101,102,103,104]
Physical modalities intervention	Neuromuscular electrical stimulation	[105]
	Extracorporeal shockwave therapy	[106]
	Photobiomodulation	[107]

* Indicate the trials derived from animal studies.

### 3.1. Nutritional Intervention

Dietary modification plays a fundamental role in the management of patients with CKD [85,86]. Controlling metabolic acidosis by adjusting dietary composition can be employed as the primary nutritional intervention. A low-protein diet (LPD) is recommended for patients with CKD stage 3–5 and metabolically stable patients by providing 0.55–0.60 g dietary protein/kg body weight/day in patients or a very low-protein diet providing 0.28–0.43 g dietary protein/kg body weight/day with keto acid/amino acid analog supplementation [87]. LPD-mediated mechanisms that slow CKD progression include reduced nitrogenous waste and decreased glomerular hyperfiltration and pressure [108,109].

Metabolic acidosis attributed to CKD is a common complication that requires routine monitoring and treatment with bicarbonate supplementation [87,110]. Correction of the acid-base balance decreases net muscle proteolysis [57] and slows the progression of CKD. Dietary modification for metabolic acidosis involves increasing low dietary acid load components (e.g., vegetables or fruits) and adopting an LPD derived from non-animal sources [85,110].

Muscle protein dynamics by unchanged protein synthesis, reduction of protein degradation and increased efficiency of recycling of amino acids deriving from protein breakdown was observed in patients underwent LPD diet [111]. Although LPDs directly provide renal protection in patients with CKD and an adaptation of protein metabolism, malnutrition and PEW should be considered owing to inadequate daily calorie intake, which is related to increased mortality and worsened renal outcomes [1]. A cross-sectional study has reported that more than 80% of patients with stage 3–5 CKD exhibit insufficient daily calorie intake after receiving routine standard dietary counseling [112]. Low plasma leucine levels are indicators of muscle wasting [112]. Calorie supplementation from fat and carbohydrates has been suggested to balance low-protein content with adequate energy against development of PEW [113]. Recently, one trial found that additional commercial oral nutritional supplements suggested by the local kidney care committee in Canada, with indications for calorie or protein needs, following individualized dietitian evaluation could improve nutritional and inflammation parameters [114].

Low vitamin D levels have been associated with low muscle strength [115] and the risk of falls [116]. A CKD mouse model revealed that parathyroid hormone (PTH)/PTH receptor signaling in adipose tissue stimulated inappropriate thermogenesis and hypermetabolism, leading to muscle wasting [8]. Hyperphosphatemia was found correlated with low handgrip strength in patients with advanced CKD [117]. Treatments targeting the mineral and bone disorders developed in CKD through vitamin D or calcium supplementation may be beneficial for improving muscle wasting. Vitamin D exerts pleiotropic effects in terms of anti-inflammatory, anti-apoptotic, cardiovascular, and antineoplastic activities [88]. Vitamin D supplementation in patients was generally supported with adequate dosage [118], despite few direct effects on muscle mass and function.

As mentioned earlier, gut dysbiosis has potential cross-links with kidney disease, and this relationship has been uncovered [34,84]. Interventions targeting the microbiota to prevent CKD progression include probiotics, prebiotics, or a combination of both. AST-120 is an adsorbent compound that removes uremic toxins when orally administered [89,90], and is potential to improve sarcopenia [91]. More recent studies have focused on the possibility of applying fecal microbiota transplantation to restore the balance of gut microbiota in diseases other than *Clostridium difficile*-associated diarrhea [119].

### 3.2. Pharmacological Therapy and Potential Therapeutic Intervention

The discovery of novel molecular pathways has facilitated the proposal of a growing number of pharmacological therapies and potential therapeutic targets. Myostatin, a signal mediated via the ActRIIB receptor in skeletal muscle, is related to muscle wasting by inhibiting protein synthesis and enhancing muscle protein degradation [120]. Anti-myostatin peptibodies reportedly reverse mouse muscle wasting and body weight loss, suppress circulating inflammatory cytokine levels, decrease protein degradation, increase protein synthesis, and facilitate satellite cell function, as well as IGF-1 intracellular signaling [92]. A previous study found that ActRII blockade using bimagrumab was related to muscle mass gain in animal models [93].

Translation of preclinical studies targeting myostatin has shown promising but heterogeneous results in human clinical trials [42]. One trial adopted a fusion protein of ActRIIB and IgG1-Fc in patients with Duchenne muscular dystrophy and showed trends of increased lean muscle mass and bone mineral density and distance maintenance during the 6-min walk test; however, the trial was discontinued owing to non-muscle-related adverse events [121]. Increased gingival bleeding, epistaxia, and talangiectasias were observed, possibly resulting from inhibition of angiogenesis by cross-react with bone morphogenic proteins (BMP). In addition, myostatin inhibitors potentially cross-react with TGF-β family members such as (activins, BMPs and growth differentiation factor 11) that share similar domain architecture, resulting in unexpected adverse events in humans [42,122]. A more precise myostatin-targeted medication is under development.

Increased levels of systemic inflammatory cytokines, such as interleukin (IL)-1, IL-6, and TNF-α, are well-documented in patients with CKD [94,123,124]. Higher baseline IL-1β levels are associated with a more rapid decline in phase angle, an indicator for muscle function, after one-year HD [125]. Recent animal studies have shown that systemically administered IL-1 blockade attenuated muscle wasting [94,95], by decreasing serum and muscle inflammation cytokines, lessening energy expenditure, and raising appetite and weight gain. Hemodiafiltration (HDF) has potential to improve chronic-inflammation by enhancing clearance of medium-high molecular weight solutes through combination of diffusion and convection. The clinical trials indicated HDF improved the inflammatory cachexia and increased bone formation in children [126,127], and reduced inflammation in adult chronic HD patients [128].

Anorexia is another crucial issue contributing to malnutrition. Ghrelin and leptin are two counter-regulatory hormones involved in appetite regulation. Increased leptin levels were observed in patients undergoing dialysis, suppressing appetite, and leading to cachexia [129,130]. Ghrelin has three circulating products: acyl-ghrelin, des-acyl ghrelin, and obestatin [131]. In patients with CKD, elevated anorexigenic forms of ghrelin (des-acyl ghrelin and obestatin) were documented without a compensatory increase in acyl-ghrelin [132]. Animal studies have shown that the administration of acyl-ghrelin could increase muscle mass and enhance muscle mitochondrial activation [96], and the application of pegylated leptin receptor antagonists slowed muscle wasting [97].

Specific miRNAs (miRNA-23a and -27a) were shown to be related to attenuate muscle wasting and improve muscle strength in mice with CKD, which are critical signaling factors in exercise-induced adaptations [72]. The selective packing of miRNAs into exosomes may be a novel pharmaceutical exercise mimetic for overcoming CKD-related muscle wasting or cachexia. Solagna et al. reported that an increase in activin A, a member of the TGF-β protein family, in experimental CKD mice led to muscle wasting [98], while pharmaceutical blockade of activin A reduced muscle wasting. Activin A, excreted from specific kidney fibroblasts and juxtaglomerular apparatus cells, can act as a pro-cachectic factor and accumulate in the plasma, owing to reduced kidney clearance and a vicious renal/muscle signaling cycle, thereby resulting in muscle wasting. This study provided an important model of kidney-muscle crosstalk [98].

### 3.3. Exercise Training

Exercise intervention is recommended in addition to pharmacological and nutritional interventions, with clinical benefits such as improved health-related quality of life and delayed muscle wasting in patients with CKD [133,134].

Both aerobic and resistance exercises performed individually or in combination have been proposed for patients with dialysis or non-dialytic CKD [99,100]. Animal models have demonstrated that resistance exercise increases insulin and IGF-1 expression and enhances the IGF-1 signaling pathway [135,136], whereas both aerobic and resistance exercise could reduce muscle proteolysis and improve phosphorylation of Akt and FoxO1 [137]. However, increased protein synthesis and mediator levels increased muscle progenitor cell number, while upregulated expression levels of *MyoD*, *myogenin*, and *embryonic myosin heavy chain* mRNAs were observed only in the resistance exercise group [137].

Traditionally, resistance exercise can reportedly target both muscle wasting and muscular dysfunction, given its anabolic effect on muscle protein metabolism [138]. In a clinical trial, resistance exercise could reduce inflammatory cytokines [101,102], but the effect on muscle mass, measured by the cross-sectional area (CSA), was inconsistent. A positive result reported significant enhancements in quadriceps CSA and volume, knee extensor strength, and exercise capacity following lower limb resistance exercise in patients with non-dialytic CKD [103], as well as a histological increase in satellite cell number in type I fibers and myonuclear content of type II fibers in patients undergoing dialysis after resistance training [104]. Conversely, studies have demonstrated discrepant results with improvements in physical function but unaltered muscle fiber size after resistance training in patients undergoing dialysis [139,140]. The optimal patient population, exercise prescription (e.g., frequency, intensity, time, and type), and combined interventions need to be explored.

### 3.4. Physical Modalities Intervention

Physical modalities are potential interventions for regaining muscle function, delaying muscle wasting, and/or restoring muscle mass. Neuromuscular electrical stimulation (NMES) has been applied in stroke rehabilitation, musculoskeletal disorders (e.g., post-anterior cruciate ligament surgery), and prolonged bedridden state due to medical illness, which causes skeletal muscle wasting [141]. A randomized controlled study found that the application of NMES during HD improved muscle strength and architecture in the vastus lateralis muscle, as determined by ultrasonography evaluation; however, functional capacity did not significantly differ from that of the control group [105].

Extracorporeal shockwave therapy (ESWT), classified into radial shockwave (RSW) and focus shockwave therapies [142], is widely employed to treat musculoskeletal diseases. A novel trial evaluating the effect of RSW on muscle mass and function in patients undergoing HD [106] revealed that a 12-week RSW treatment significantly increased appendicular skeletal muscle mass and improved muscle function, as measured by timed up-and-go and sit-to-stand tests. Although the mechanism of ESWT is poorly understood, previous studies have shown that ESWT induces angiogenesis and reduces inflammation in soft and hard tissues [143,144,145], which may be linked to downstream muscle functional restoration.

Photobiomodulation (PBM) therapy using a low-level laser can be employed as a potential alternative therapy to target muscle wasting in patients with dialytic CKD. A recent randomized controlled clinical trial revealed an increase in lower limb muscle strength and functional capacity in patients who received PBM over the bilateral lower limb [107]. The mechanisms underlying PBM include increased enzymatic activity of cytochrome c oxidase [146], improved microcirculation, and, consequently, elevated ATP production from muscle mitochondria, which in general can overcome the mitochondrial dysfunction observed in patients with CKD.

It is worthwhile to mention that physical modalities pertain to passive interventions rather than active motion from exercise. Exercise-induced multi-organ adaptations lead to improvements in physical function and health-related quality of life. The indications for application of these modalities should not be overemphasized, and its role may be complementary for potential additional benefit from standard medical care.

## 4. Conclusions

A complex pathomechanism mediates muscle wasting in patients with CKD. Metabolic derangements following reduced kidney function induce an imbalance in muscle protein homeostasis. Molecular pathways in response to CKD, including suppressed protein synthesis, increased protein degradation, and impaired muscle regeneration, have been discovered. Certain molecular mechanisms of kidney-muscle crosstalk have also been proposed and might afford potential therapeutic targets. As growing studies unravel the pathway leading to muscle wasting, a greater number of novel interventions are expected to overcome muscle loss. Exercise and nutritional interventions can be included in standard medical care, whereas physical modalities to enhance muscle function and restore muscle mass are warranted. Potential pharmaceutical targets, such as microRNA packing by exosomes and activin A blockade, could be expected in the future.

## Figures and Tables

**Figure 1 ijms-23-06047-f001:**
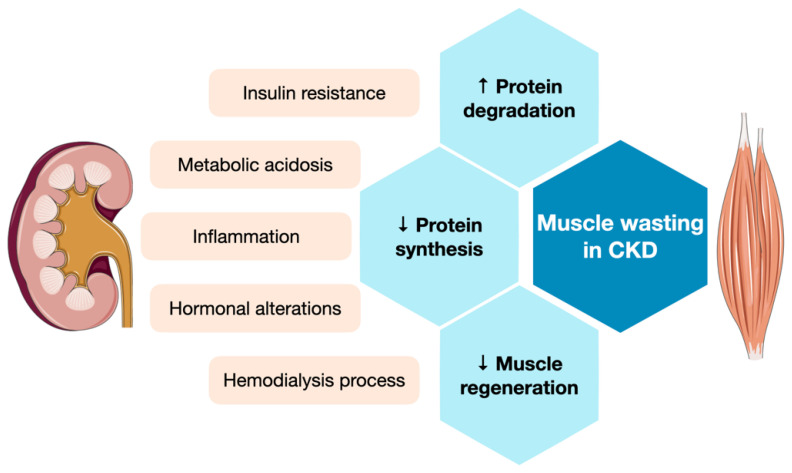
Illustration of the relationship between CKD-related factors and muscle wasting. Upward arrows indicate increase or activation; downward arrows indicate decrease or suppression. The graphics used in this figure are adapted from Sevier Medical Art (www.smart.servier.com, accessed on 5 April 2022).

**Figure 2 ijms-23-06047-f002:**
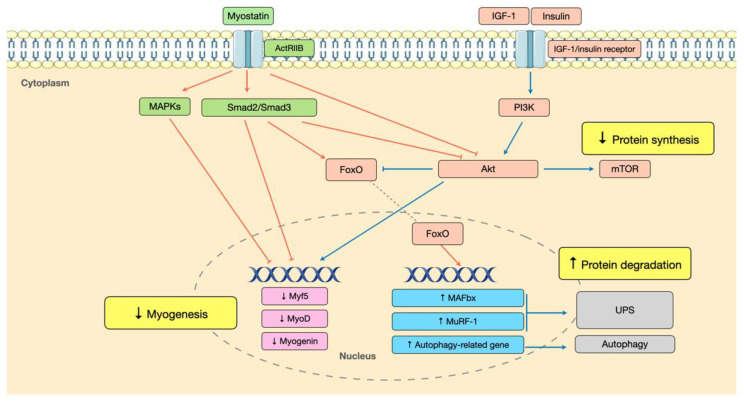
Altered cellular signaling pathways underlying muscle wasting in CKD. (Blue lines/arrows indicate activation; red lines/arrows indicate inactivation; upward arrows indicate upregulation or increase; downward arrows indicate downregulation or decrease.) The graphics used in this figure are adapted from Sevier Medical Art (www.smart.servier.com, accessed on 5 April 2022).

## Data Availability

Not applicable.

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
