# Peer review of "Muscle Wasting in Chronic Kidney Disease: Mechanism and Clinical Implications—A Narrative Review"

_ijms, 2022, doi:10.3390/ijms23116047_

Round 1

Reviewer 1 Report

The authors presented a detailed overview of muscle wasting in chronic kidney disease. I have some comments which may improve the quality of the manuscript:

In the Introduction section:

  1. The authors need to discuss in brief the protein energy wasting and frailty, both of these conditions are strongly correlated to muscle wasting, and affect quality of life, mortality and overall morbidity.
  2. Moreover, in the lines 32-36, bone mineral imbalance (vitamin D deficiency, secondary hyperparathyroidism) should be included in the list of possible contributing factors of muscle wasting and the above mentioned articles need to be added in the references of this sentence.
  3. In lines 73-76, the authors discuss the role of IGF-1/myostatin imbalance in the development of muscle wasting. The clinical impact of these parameters as biomarkers of muscle wasting need to be described in the manuscript.
  4. In the nutritional intervention, in line 333 the authors describe the link between vitamin D and muscle strength. There should be a sentence regarding the emerging role of secondary hyperparathyroidism on muscle wasting, suggested by both in vivo and clinical studies, and that treatment with calcitriol and calcium may be beneficial for muscle status..
  5. In the therapeutic interventions the authors need to add the low-inflammatory impact dialysis by hemodifiltration, which has been proven beneficial for preservation of muscle status in clinical studies.

Reviewer 2 Report

I considered the manuscript entitled “Muscle Wasting in Chronic Kidney Disease: Mechanism and Clinical Implications – A Narrative Review” by Tsai-Chin Cheng, et al

that is intended to be published in IJMS journal.

The manuscript is in the average. It offers interesting data explaining the mechanisms of muscle homeostasis but the Chapter of interventions should be ameliorated and more data added focusing in CKD patients.

Line 56: transient and return to baseline status post-hemodialysis [, what means post hemodialysis?? After one dialysis session?

Concerning low protein diet, authors do not inform about the measure in terms of muscle protection. It is just a measure of renal protection, but it appears as dangerous for muscle protection. It is controversial. Add more information than: additional commercial oral nutritional supplements

The lack of measures to protect the muscle in CKD is really discouraging

Myostatin inhibitors potentially cross-react with TGF-β family members, resulting in unexpected adverse events in humans. Could you explain better?

Recent animal studies have shown that systemically administered interleukin (IL)-1 blockade attenuated muscle wasting. Discuss mechanisms or findings from the study

The Chapter Therapeutic Strategies for Muscle Wasting is scarce of information, should you add additional information really focused in muscle

Round 2

Reviewer 2 Report

No further comments